# Addressing Drug Resistance in Cancer: A Team Medicine Approach

**DOI:** 10.3390/jcm11195701

**Published:** 2022-09-27

**Authors:** Prakash Kulkarni, Atish Mohanty, Supriyo Bhattacharya, Sharad Singhal, Linlin Guo, Sravani Ramisetty, Tamara Mirzapoiazova, Bolot Mambetsariev, Sandeep Mittan, Jyoti Malhotra, Naveen Gupta, Pauline Kim, Razmig Babikian, Swapnil Rajurkar, Shanmuga Subbiah, Tingting Tan, Danny Nguyen, Amartej Merla, Sudarsan V. Kollimuttathuillam, Tanyanika Phillips, Peter Baik, Bradford Tan, Pankaj Vashi, Sagun Shrestha, Benjamin Leach, Ruchi Garg, Patricia L. Rich, F. Marc Stewart, Evan Pisick, Ravi Salgia

**Affiliations:** 1Department of Medical Oncology and Therapeutics Research, City of Hope National Medical Center, Duarte, CA 91010, USA; 2Department of Systems Biology, City of Hope National Medical Center, Duarte, CA 91010, USA; 3Integrative Genomics Core, City of Hope National Medical Center, Duarte, CA 91010, USA; 4Montefiore Medical Center, The University Hospital for Albert Einstein College of Medicine, Bronx, NY 10467, USA; 5Department of Medical Oncology and Therapeutics Research, City of Hope National Medical Center, 1000 FivePoint, Irvine, CA 92618, USA; 6Department of Medical Oncology and Therapeutics Research, City of Hope National Medical Center, 1100 San Bernardino Road, Suite 1100, Upland, CA 91786, USA; 7Department of Pharmacy, City of Hope National Medical Center, Duarte, CA 91010, USA; 8Department of Medical Oncology and Therapeutics Research, City of Hope National Medical Center, 1250 S. Sunset Ave., Suite 303, West Covina, CA 91790, USA; 9Department of Medical Oncology and Therapeutics Research, City of Hope National Medical Center, 1601 Avocado Ave., Newport Beach, CA 92660, USA; 10Department of Medical Oncology and Therapeutics Research, City of Hope National Medical Center, 19671 Beach Blvd. #315, Huntington Beach, CA 92648, USA; 11Department of Medical Oncology and Therapeutics Research, City of Hope National Medical Center, 38660 Medical Center Dr, Suite A380, Palmdale, CA 93551, USA; 12Department of Medical Oncology and Therapeutics Research, City of Hope National Medical Center, 16300 Sand Canyon Ave., Suite 207, Irvine, CA 92618, USA; 13Department of Medical Oncology and Therapeutics Research, City of Hope National Medical Center, 44151 15th St. West, Lancaster, CA 93534, USA; 14Cancer Treatment Centers of America, CTCA Chicago, 2520 Elisha Avenue, Zion, IL 60099, USA; 15Cancer Treatment Centers of America, CTCA Phoenix, 14200 West Celebrate Life Way, Goodyear, AZ 85338, USA; 16Department of Medical Oncology and Therapeutics Research, City of Hope National Medical Center, 15031 Rinaldi St., Suite 150, Mission Hills, CA 91345, USA; 17Cancer Treatment Centers of America, CTCA Atlanta, 600 Celebrate Life Parkway, Newnan, GA 30265, USA

**Keywords:** drug resistance, drug tolerance, eco-evolutionary, intermittent therapy, adaptive therapy, continuous therapy, Team Medicine, intrinsically disordered proteins

## Abstract

Drug resistance remains one of the major impediments to treating cancer. Although many patients respond well initially, resistance to therapy typically ensues. Several confounding factors appear to contribute to this challenge. Here, we first discuss some of the challenges associated with drug resistance. We then discuss how a ‘Team Medicine’ approach, involving an interdisciplinary team of basic scientists working together with clinicians, has uncovered new therapeutic strategies. These strategies, referred to as intermittent or ‘adaptive’ therapy, which are based on eco-evolutionary principles, have met with remarkable success in potentially precluding or delaying the emergence of drug resistance in several cancers. Incorporating such treatment strategies into clinical protocols could potentially enhance the precision of delivering personalized medicine to patients. Furthermore, reaching out to patients in the network of hospitals affiliated with leading academic centers could help them benefit from such innovative treatment options. Finally, lowering the dose of the drug and its frequency (because of intermittent rather than continuous therapy) can also have a significant impact on lowering the toxicity and undesirable side effects of the drugs while lowering the financial burden carried by the patient and insurance providers.

## 1. Introduction

Cancer is a major healthcare crisis and one of the leading causes of death in the world. In 2020 alone, worldwide, there were ~18 million cancer cases, and cancer accounted for nearly 10 million deaths, or nearly one in six deaths, in the same year, Sung et al., 2020 [1]. In less than two decades from now (by 2040), the number of new cancer cases per year is expected to rise to >27 million and the number of cancer-related deaths to >16 million [2]. These whopping numbers underscore the magnitude of the health care crisis and the economic burden of this burgeoning problem across the globe. Although the disease presents several challenges depending on the cancer type and, in some cases, ethnicity, the emergence of drug resistance remains a major common concern in treating all cancer patients.

Several confounding factors appear to be contributing to this challenge. Here, we first discuss some of the challenges associated with drug resistance in cancer. We then discuss how a ‘Team Medicine’ approach [3], involving an interdisciplinary team of scientists with expertise in physics, biophysics, mathematics, evolutionary biology, bioinformatics, data science, computational biology, and cancer biology, working with clinicians, has provided new opportunities and new therapeutic strategies that could potentially preclude or delay the emergence of drug resistance in several cancers. We conclude by proposing a few innovations to our approach in treating cancer: (1) the novel therapeutic strategies such as intermittent drug treatment at moderate dosage as opposed to continuous treatment at high dosage and (2) leveraging precise knowledge of the tumor phenotypic landscape in designing personalized therapy through deeper consideration of genetic, epigenetic, and transcriptomic information.

## 2. Is Drug Resistance Genetic or, Are Non-Genetic Mechanisms Involved?

For well over a century, since Theodore Boveri’s ground-breaking observations in the early 1900s (*Zur Frage der Entstehung maligner Tumouren*), cancer has been thought to be a genetic disease [4,5,6,7]. In fact, today, a genetic basis underlying the origin of cancer, its progression through distant metastasis, and the emergence of drug resistance is practically common knowledge. The following excerpt from an influential review in Nature Medicine by Vogelstein and Kinzler [8], titled “Cancer genes and the pathways they control”, underscores the prevailing ethos: “*The revolution in cancer research can be summed up in a single sentence: cancer is, in essence, a genetic disease. In the last decade, many important genes responsible for the genesis of various cancers have been discovered, their mutations precisely identified, and the pathways through which they act characterized*.” (Our bold for emphasis). Pursuant to this landmark review, a decade later, Vogelstein and Kinzler [9] further emphasized the genetic nature of cancer in a perspective article, “The path to cancer—three strikes and you’re out”, in the leading medical journal, New England Journal of Medicine. The authors wrote “*Focusing on driver-gene mutations and the pathways they control has rendered complex cancer-genome landscapes intelligible. In solid tumors of adults, alterations in as few as three driver genes appear to suffice for a cell to evolve into an advanced cancer.*” (Our bold is for emphasis).

Highly influential articles like these and countless others have helped to firmly establish the genetic basis of cancer and, much like the modern synthesis in evolutionary theory [10], provide a conceptual framework to understand cancer and its link to Darwinian evolution. Furthermore, specific mutations have been leveraged as hallmarks for a conclusive diagnosis and the staging of specific cancer types, and highly potent drugs that specifically bind to the mutant target oncoproteins have been developed, adding to the precision with which individual patients are treated [11,12,13].

However, contrary to the prevailing wisdom, emerging evidence indicates that mechanisms such as epigenetic modifications and protein interaction network (PIN) rewiring—defined as changes in the interaction of individual proteins in signaling pathways in response to environmental changes—can also contribute to various aspects of cancer, including its origin, progression, and the emergence of drug resistance [14,15,16,17,18,19,20], underscoring the role of non-genetic mechanisms. It is now evident that both non-genetic and genetic mechanisms are involved, especially in acquired drug resistance and, furthermore, as discussed below, the irreversible resistance to a drug that can be acquired via an intermediate, reversible tolerant state via non-genetic mechanisms.

## 3. Discerning Drug Tolerance and Resistance

Perhaps one of the main reasons for our failure to overcome drug resistance in cancer, whether innate or acquired, may have to do with the difficulty in how we perceive the phenomenon. Unfortunately, and erroneously (even if inadvertently), it is assumed that drug resistance, tolerance, and persistence are synonymous or equivalent and hence, are used indiscriminately [21]. However, these are quite distinct and nuanced phenomena, as elegantly demonstrated in microbiology.

In bacteria for example, resistance is defined as the ability of an organism to grow at high concentrations within the presence of a drug. Resistance is typically due to mutations and is heritable transgenerationally. On the other hand, tolerance is more generally used to describe the ability, whether inherited or not, to survive transient exposure to high concentrations of a drug, and persistence is defined as the ability of a subpopulation to survive long-term exposure to high concentrations of a drug. Persistence is typically observed when the majority of the population is rapidly killed following drug treatment while a subpopulation persists for a much longer period of time [22,23] (Figure 1). Since rigorous definitions are lacking in the cancer field, the term ‘resistance’ remains ambiguous or confusing at best; thus, this further adds to the difficulty of defining a patient’s response to a drug. For example, if a patient does not respond to a drug, is the patient’s tumor tolerant and hence potentially reversible? or is it truly resistant and hence irreversible? Perhaps, publicly available databases dedicated to cancer drug resistance could help alleviate some of the confusion [24].

## 4. Current Treatment Strategies May Be Counterproductive

To further complicate the issue, in addition to being taken for granted that cancer is a genetic disease, it is also believed that cancers arise by Darwinian evolution, involving a reiterative process of clonal expansion, genetic diversification, and selection within the adaptive landscapes of the tissue ecosystems they inhabit [26]. Therefore, while therapeutic intervention can destroy cancer clones and erode their habitats, the same intervention, especially when administered continuously, may also inadvertently provide a potent selective pressure for the expansion of drug-resistant phenotypes. However, recent developments in sequencing and omics technologies, coupled with theoretical advancements, have provided an expanded understanding of the cancer phenotypic landscape. They highlight non-genetic mechanisms that enable cancer cells to reversibly adapt to their environment, unlike genetic mechanisms that are irreversible, underscoring the dire need to reconcile the two mechanisms. Additionally, plastic phenotypes such as cancer stem cells (CSCs) are also well-recognized entities contributing to drug resistance in many cancers [27,28,29]. Therefore, the dynamic heterogeneity and the dynamic transitions between CSCs and non-CSCs and their significance in metastasis and drug resistance warrant a deeper understanding of the underlying mechanisms.

Indeed, a recent article by Sui Huang [30] in a Special Issue of *Trends in Cancer* entitled, ‘Quantitative Cancer Biology’, further emphasizes the need to reconcile non-genetic plasticity with somatic evolution in cancer: “Posttreatment progression of tumors is commonly explained by somatic Darwinian evolution (i.e., selection of cells carrying genetic mutations that create more aggressive cell traits). But cancer genome and transcriptome analyses now paint a picture far more complex, prompting us to see beyond the Darwinian scheme: non-genetic cell phenotype plasticity explained by alternative stable gene expression states (‘attractors’), may also produce aggressive phenotypes that can be selected for, without mutations. Worse, treatment may even induce cell state transitions into more malignant attractors.” (Our bold for emphasis).

## 5. Emergence of Irreversible Drug Resistance via a Potentially Reversible Tolerant State

Because they exhibit a high degree of phenotypic plasticity [31], cancer cells can switch on cell-autonomous traits such as persistence and quorum sensing when stressed [32]. Cancer cells exhibiting a persister trait are slow growing and can give rise to tolerant cells that, as discussed above, play an important role in the emergence of true drug resistance [33,34]. To comprehend how drug resistance may evolve from an intermediate tolerant state, it is helpful to view cancer from Waddington’s [35] epigenetic landscape perspective (Figure 2). The concept of a “landscape” represents a high-dimensional state space where each phenotype acts as an “attractor” determined by the underlying PIN and is buffered against environmental fluctuations. Cellular PINs are organized following scale-free (rather than random) configurations. Therefore, PINs follow a power law distribution, wherein a few nodes (referred to as hubs) have numerous edges (connections), while most nodes have few or very few edges. Scale-free networks are resilient to random node failures; however, they are susceptible to targeted attacks on hubs. PINs serve as the main conduit of information flow with crucial roles in cellular decision-making [17,36]. PINs can determine the fate that a cell can realize and can robustly establish its phenotype because they are minimally frustrated [37]. Frustration is defined as the inability of the system to simultaneously minimize the competing interaction energies between its components [38]. In cancer cells, PIN frustration can be a viable mechanism of achieving phenotypic plasticity besides epigenetic changes. As discussed in the next paragraph, PIN frustration is driven by a special class of proteins with high structural flexibility and an ability to interact with multiple partners.

Intrinsically disordered proteins (IDPs) are proteins that lack a rigid 3D structure and exist as ensembles of interconverting conformations [40]. Because they are highly malleable, IDPs interact with multiple partners and thus occupy hub positions in PINs. Noise contributed in part by the conformational dynamics of IDPs (‘conformational noise’) plays an important role in cell-fate specification [17,18,36]. Nonetheless, in response to stress, especially in conjunction with post-translational modifications such as phosphorylation, IDPs engage in promiscuous interactions and drive phenotypic transitions by PIN rewiring [17,18,36,41]. This heuristic can uncover new attractors in the system—including “cancer attractors,” defined as hidden stable states of PINs [42,43]—and cause phenotypic switching. Upon stress withdrawal, IDPs reconfigure the PIN to return to the original phenotype, highlighting the reversible nature of phenotypic switching. However, if stressful conditions persist, chronic stress can result in persistent network frustration, which is relieved by acquiring specific DNA mutations and/or other genetic alterations, making the phenotypic change permanent [32,44]. Thus, it follows that, non-genetic mechanisms can eventually lead to acquired resistance via irreversible genetic changes at the individual cell level. Further, as discussed below, this stepwise trajectory to drug resistance also highlights an unprecedented opportunity to preclude or delay it by controlling stress (by manipulating drug dose/time) experienced by the tolerant tumor cells.

## 6. Intermittent or ‘Adaptive’ Therapy—An Eco-Evolutionary Principles-Based Therapeutic Strategy to Preclude or Delay Onset of Drug Resistance

The rationale for this treatment strategy is based on the principles of ecology and evolution. Within the tumor microenvironment (TME), cancer cells reside with several other cell types that cohabit in this space. By producing growth factors and proinflammatory cytokines to promote angiogenesis, these cells create an ecosystem that enables the malignant cell population to grow and flourish. Therefore, group behavior, an emergent property defined as the collective actions performed by the individuals in the group as a whole, imposes costs and benefits to the participating individuals that can be recast as a game pay-off matrix. Thus, evolutionary game theory, which provides an elegant conceptual framework to capture the frequency-dependent nature of ecosystem dynamics, can be used to model tumor progression and dynamics. In fact, game theory can also be leveraged to discern the games cancer cells play by either cooperating or competing in the absence or presence of stress (selective pressure). Therefore, treatment options that consider the strategies cancer cells adopt to deal with drug effects have been developed and are referred to as intermittent or ‘adaptive’ therapy [45].

Typically, such treatment protocols call for initial therapies to induce adaptive changes in the tumor environment such that the proliferation of resistant clones is markedly suppressed for extended periods. In this paradigm, it is recommended that therapy is administered in small doses to attenuate tumor growth but has just enough to improve the symptoms. In other words, it is recommended that a minimal dose of the treatment (that is necessary and not at the maximum tolerated) must be used to achieve the desired result. Furthermore, treatment is administered intermittently (in alternate cycles) rather than continuously (given at every scheduled time) so that a drug-sensitive tumor population will be sustained at the expense of the resistant ones. In addition, drug combinations/epigenetic modifiers may be used in the intermittent/adaptive therapy regimen if necessary. Although, in this treatment strategy, the tumor is not completely eradicated, and it is likely that the tumor progresses between treatments; it is also likely that the tumor cells will continue to be sensitive to therapy and therefore delay or may even preclude the onset of drug-resistant disease (Figure 3) and thus, prolong overall survival.

Indeed, intermittent therapy for prostate and breast cancer [46], melanoma [47], rectal cancer [48], and pediatric sarcomas [49] are currently being evaluated in the clinic with promising results. In some other cancers, e.g., non-small cell lung cancer, our preclinical data obtained using a Team Medicine approach also indicate that such strategies may prove successful as well [50]. They could also have a significant impact on mutant KRAS inhibitors, such as sotorasib, which was recently approved for lung cancer treatment and is promising but is already being reported to develop resistance [51,52,53,54,55,56].

Despite the promise and initial success, more research is warranted to gain a deeper understanding of intermittent/adaptive therapy and of the side effects, if any, especially when drug combinations are to be used. For example, in one study, it was reported that when a tumor is sensitive to two or more drugs, the simultaneous application of these drugs could result in the emergence of cells resistant to both therapies. However, when these drugs were applied one at a time, a subpopulation of cells was sensitive to one or the other drug, delaying the emergence of double-resistant cell clones [57]. Conversely, in another study on lung cancer, it was observed that concurrent targeting of multiple kinases, rather than a single kinase, resulted in the complete (100%) inhibition of tumor growth. The latter strategy was only effective when intermittent and not continuous therapy was administered. One possibility for this dramatic inhibition is likely due to the lack of adaptability of the tumor cells to the changing fitness threshold imposed by selection [58].

Two randomized trials have investigated intermittent dosing regimens with BRAF and MEK inhibitors for the treatment of BRAF-mutated advanced malignant melanoma. Gonzalez-Cao et al. [59] reported lower median progression-free survival (PFS; 6.9 months versus 16.2 months; *p* = 0.079) with the intermittent use of vemurafenib and cobimetinib when compared to continuous dosing in 70 patients with treatment-naïve advanced melanoma. No statistical difference was observed for overall survival (OS) or in objective response rates (OSS). In another randomized, open-label, phase two trial, comparing continuous versus intermittent BRAF and MEK inhibition in patients with BRAF-mutated melanoma, Algazi et al. [60] reported that continuous dosing was associated with a statistically significant improvement in median PFS compared with intermittent dosing (9.0 months versus 5.5 months, *p* = 0.064, pre-specified two-sided α = 0.2). Even though there was a PFS difference between the two groups, no differences were observed in the OS and ORR. This could possibly be due to the finding that intermittent dosing was associated with longer survival after progression (HR 0.76; 80% CI 0.78 to 1.00). Maio et al. [61] improved efficacy with intermittent MEK inhibition when combined with anti-PD-1 immunotherapy (pembrolizumab) in patients with advanced or metastatic BRAF-mutated solid tumors (36% colorectal cancer and 10% melanoma). ORR was reported to be 8% effective with concurrent and 28% effective with the intermittent dosing groups, respectively. Several trials have investigated the role of intermittent androgen deprivation therapy (ADT) in the treatment of advanced prostate cancer. These trials have reported that intermittent ADT has similar clinical outcomes when compared to continuous ADT with no statistically significant differences in OS, cancer-specific survival, or PFS [62,63,64]. However, intermittent ADT is associated with an improved quality of life and a lower risk of adverse events [63,64,65]. Ongoing trials are now investigating intermittent ADT in combination with additional therapies, such as radiation or immunotherapy, that can potentially further increase the time of systemic treatment [66]. Thus, the clinical trials so far, which have investigated intermittent dosing regimens, have yielded mixed results, highlighting the complexity of translation preclinical studies into human trials and the challenges of selecting the optimal dosing regimen. Future trials should focus on exploring this approach in biomarker-selected populations, as well as elucidating which subgroups of patients may benefit most from this approach.

Since intermittent therapy relies on drug-sensitive cells to suppress the proliferation of the tolerant cells, the success of such therapies is dependent on the initial population of the two cell types within the tumor. The hypothetical scenario shown in Figure 4, where the three models represent the three patients, may help better appreciate the underlying nuances. Here, Patient 1, with the highest number of sensitive cells compared to tolerant cells, will have more prolonged progression-free survival in response to drug treatment. The second patient with an equal number of sensitive and tolerant populations will have shorter progression-free survival, while Patient 3, with the highest number of tolerant cells, will have the shortest progression-free survival. The purpose of intermittent therapy is to prevent the development of drug refractory-resistant clones and is based on the presumption that the sensitive cells, in the absence of the drug, will compete with resistant cells and grow faster to suppress the growth of resistant cells. Thus, the more percentage of sensitive cells, the better the response to the intermittent therapy, and, thus, in scenarios like model 1 and model 2, the success rate will be higher compared to model 3. Moreover, the inherent assumption in intermittent therapy, that the sensitive and tolerant cell types are competitive, may not hold universally. Depending on the tissue/cancer type, more complex ecological relationships may exist among the different cell types (such as cooperation or competition, depending on the stress level/drug dosage) that may require more fine-tuned, dynamic adjustments to drug schedules/dosages as the therapy progresses and the tumor phenotypic landscape evolves.

Thus, it follows that, before initiating a therapeutic strategy, a detailed genetic (and possibly transcriptomic and epigenetic) analysis of the patient’s tumor is imperative. For example (Figure 5), patients with an RAF mutation are likely to respond better to BRAF inhibitors, while patients with BRAF and MEK mutations may respond less to the same inhibitors, and patients with MEK and PI3K mutations will not respond to BRAF inhibitors at all. Likewise, patients with activated RTK signaling are unlikely to respond to the BRAF inhibitor drug treatment effectively, as the tumor will likely take advantage of the bypass signaling through the AKT -mTOR pathway to overcome the drug effect. Therefore, pre-trial validation of the mutations in the tumor by NGS and the expression of the activated signaling need to be determined so that patients with a similar mutational profile or expression status may be grouped into a specific cohort. As illustrated in Figure 5, patients in each of the four cohorts can be treated with continuous or intermittent therapy, and the statistical significance can be derived to determine the best therapeutic approach. However, comparing intermittent versus continuous treatment between the two cohorts will give insignificant information. Figure 5 also suggests the role of Team Medicine, where basic help from scientists can identify those signaling pathways that need to be targeted for effective therapy through experiments, clinicians and bioinformaticians can help to validate the basic study by analyzing thousands of public data and help to look for specific mutations that can contribute to the pathways, molecular pathologists can help in determining the expression of these proteins in the tumor biopsies, and finally, clinicians, aided by cumulative information and precise mathematical models, can design the drug treatment strategy.

## 7. Concluding Remarks

There exist multiple mechanisms that regulate phenotypic switching and drug resistance, even within a given cancer type. Furthermore, although promising in some cases, challenges still remain with regard to intermittent therapy, as discussed above. Thus, a better understanding of the mechanisms can help us to design the most effective therapeutic approach. Nonetheless, from the foregoing, it is obvious that these exciting developments in medical oncology, which expound the virtues of modern translational research, can only be made possible by a true Team Medicine approach, including basic scientists and clinicians. By incorporating treatment strategies based on the principles of ecology and evolution in clinical protocols, and by reaching out to patients who frequent those hospitals that are part of the network formed by academic centers rather than the academic centers themselves, we can enhance the precision in which we deliver personalized medicine to all our patients, regardless of their economic status or their ability to access advanced medical centers. We trust that our integrated efforts at the City of Hope, in conjunction with the cancer treatment centers of America, shall serve as a good example to those who wish to adopt this paradigm. Last but not least, lowering the dose of the drug and its frequency (because of intermittent rather than continuous therapy) can also have a significant impact on lowering the toxicity and undesirable side effects of the drugs while lowering the financial burden carried by the patient and insurance providers [67].

## Figures and Tables

**Figure 1 jcm-11-05701-f001:**
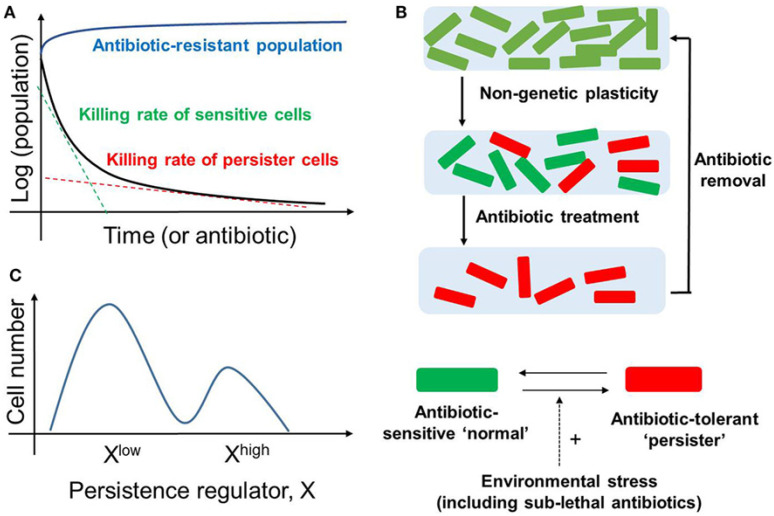
Bacterial Persistence. (**A**) Biphasic time-kill curve in bacterial populations exposed to antibiotics: faster killing rate of sensitive cell (green dotted line) followed by a slower killing rate (red dotted line) of the persisters. In contrast, the antibiotic-resistant population continues to grow in the presence of antibiotic (blue curve). (**B**) (top) An isogenic population of antibiotic sensitive cells can give rise to persisters via non-genetic/phenotypic plasticity. These slow cycling persisters survive in the antibiotic treatment and tend to resume growth and generate a new population identical to the original population upon antibiotic removal (bottom). Persisters and non-persisters can switch among one another; the switching rate can be influenced by external stress factors. (**C**) Non-genetic heterogeneity of a key regulator of persistence (say X) in an isogenic population may give rise to two (or more) subpopulations that may continue switching stochastically among themselves to maintain persistence [25].

**Figure 2 jcm-11-05701-f002:**
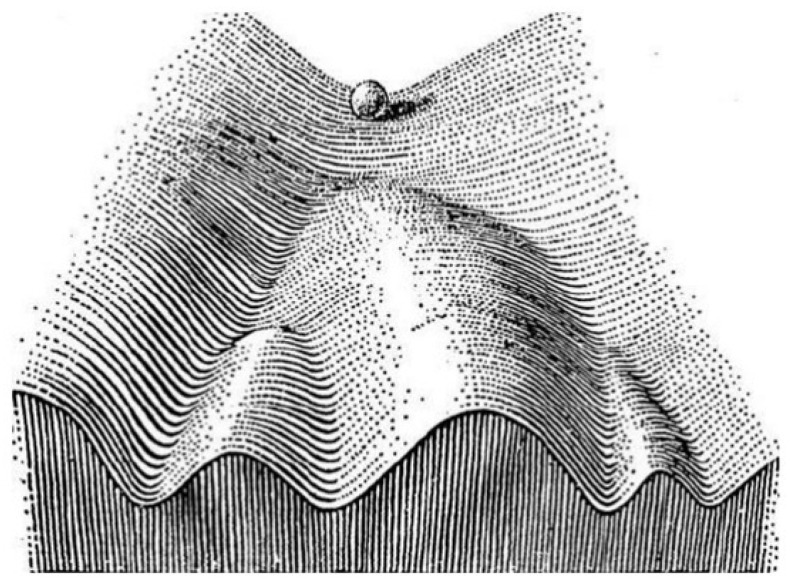
Schematic illustration of Waddington’s epigenetic landscape [35]. The ball rolling down the hill (the x axis) represents a pluripotent cell that differentiates as it rolls down the valleys. The fate of the cell is decided by the attractors that reside at the bottom of the hill (the y axis). The valleys are separated by ridges that preclude transdifferentiation [39].

**Figure 3 jcm-11-05701-f003:**
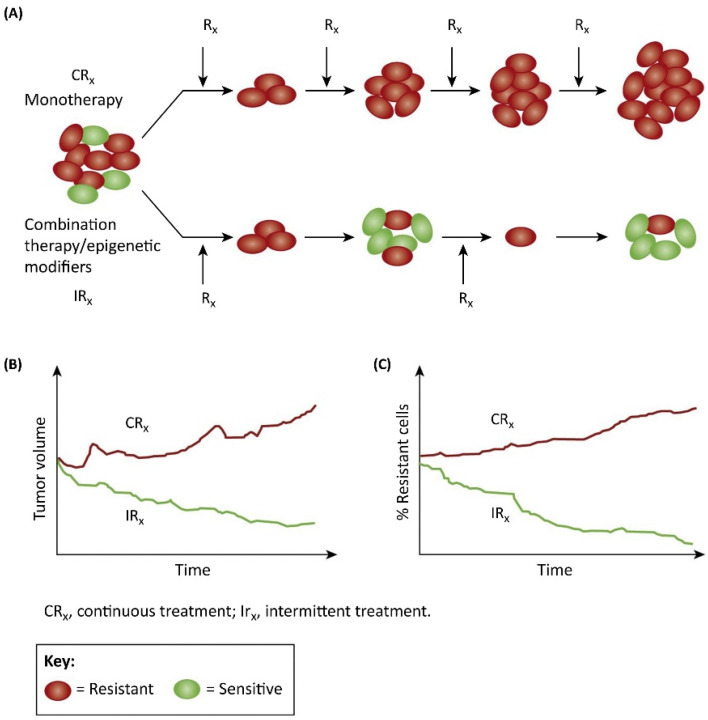
Continuous Monotherapy versus Intermittent Combination Therapy. (**A**) In continuous monotherapy, the idea is to eradicate the tumor as quickly as possible. However, this strategy can give rise to resistance, and resistant cells are expected to propagate over time (top). By contrast, combination therapy applied intermittently (bottom) could induce ‘adaptive strategies’ to change the tumor environment in such a way that the proliferation of the resistant clones can be suppressed for prolonged periods of time. Therapy is applied in small doses to reduce the tumor population only sufficient enough to improve the symptoms. Furthermore, treatment is intermittent so that drug-sensitive cells will proliferate at the expense of the resistant ones. (**B**,**C**) Although the tumor will increase in size between treatments, the extant tumor cells will continue to be sensitive to therapy [21].

**Figure 4 jcm-11-05701-f004:**
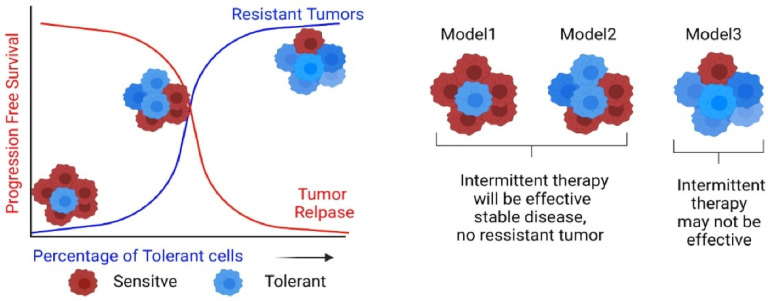
The cartoon representing the importance of tumor heterogeneity on therapeutic approach, continuous verses intermittent.

**Figure 5 jcm-11-05701-f005:**
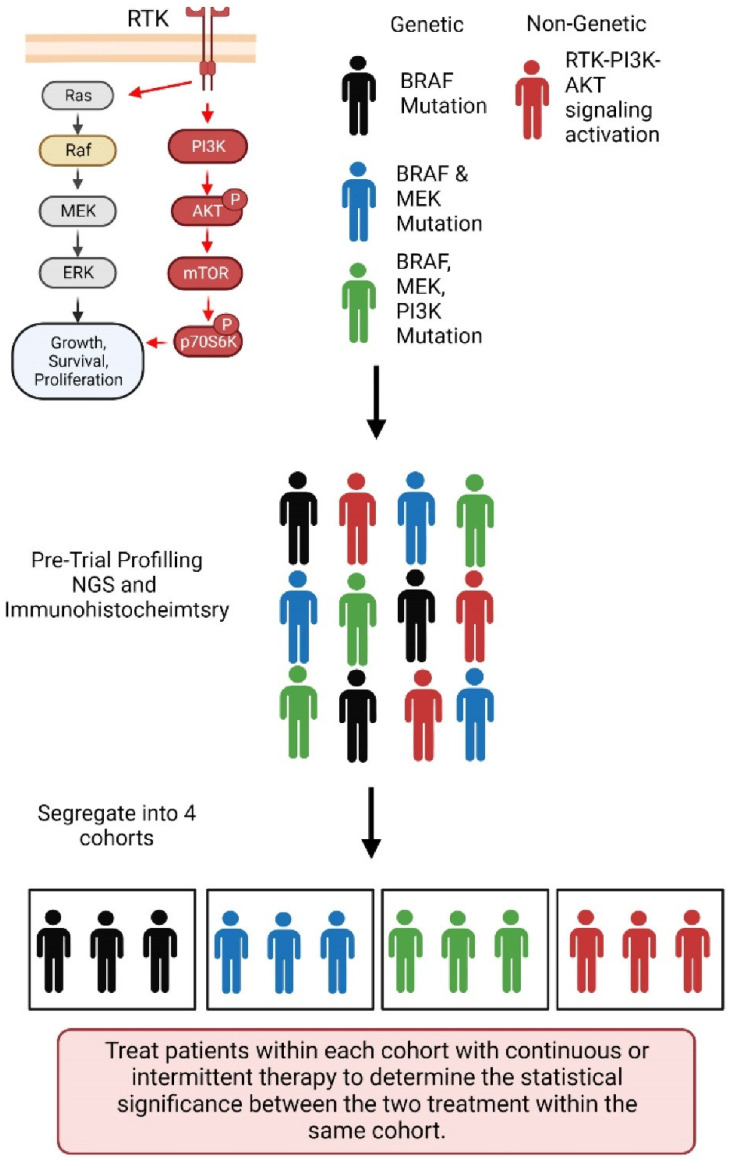
A schematic representing the pretreatment preparation for choosing the best treatment strategy. RTK, receptor tyrosine kinase; Ras, Ras protooncogene; Raf. Raf protooncogene, serine/threonine kinase; MEK, MAP kinase-ERK kinase; ERK, extracellular regulated MAP kinase; PI3K, phosphatidylinositol 3-kinase; AKT, AKT serine/threonine kinase 1; mTOR, Mechanistic Target Of Rapamycin Kinase; BRAF, B-Raf Proto-Oncogene, Serine/Threonine Kinase; NGS, next generation sequencing.

## Data Availability

Not applicable.

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
