# Peer review of "Addressing Drug Resistance in Cancer: A Team Medicine Approach"

_jcm, 2022, doi:10.3390/jcm11195701_

Round 1

Reviewer 1 Report

The manuscript entitled "Addressing Drug Resistance in Cancer: A Team Medicine Approach", discusses the problem of drug resistance and how to potentially overcome this challenge using various strategies such as intermittent or 'adaptive' therapy which are based on  eco-evolution principles.

The manuscript is written clearly with a good work flow. It is comprehensive and  contributes well to this interesting and underdeveloped field.

Due to the fact that this is a clinically focused journal, I suggest the reviewers expand on the therapeutic effects and clinical data of of intermittent dosing as several previous clinical studies have concluded opposing results i.e. intermittent dosing does not provide improved therapeutic outcomes as opposed to maximal dosing. These studies do not support intermittent dosing and show disappointing results. These studies include: 'Continuous versus intermittent BRAF and MEK inhibition in patients with BRAF-mutate melanoma: a randomized phase 2 trial' by Algazi et al. and 'Intermittent BRAF inhibition in advanced BRAF mutated melanoma results of a phase II randomized trial by Gonzalez-Cao et al. as well as several other recent clinical studies.

Furthermore, authors should add more clinical data in the form of PFS and OS correlation for relevancy. 

The authors should add further details regarding treatment setting i.e. adjuvant, palliative treatment? As what is stated in the article is not suitable for neo-adjuvant treatment.

Author Response

Author’s response: We appreciate the kind words the Reviewer has regarding the manuscript. We also wish to thank the Reviewer for the constructive comments and bringing to our attention the two very important papers by Algazi et al (Nature Med, 2020) and by Gonzalez-Cao et al (Nature Communications, 2021). We have now added these two trials to the discussion in the manuscript as well as few additional studies reporting results from trials exploring intermittent dosing schedules. Clinical outcomes including OS, PFS and ORR these studies have been added to provide clinical context. The setting for almost all trials discussed is for advanced or metastatic disease and this has also been added to the text where these trials are discussed. We have also pointed out the lingering issues with intermittent therapy so that the reader gets an unbiased view of the potential and limitation of this new treatment approach. The new text has been highlighted to draw the reviewer’s attention.

Reviewer 2 Report

1.      The introduction and conclusion of the article do not agree with each other. The problem identified in the introduction of the review is not fully discussed in the course of the article. The conclusion does not summarize the actual content of the work. In general, in the article I would like to see more facts that are contradictory, and then a summary conclusion of the author of the review.

2.      The section “3. Discerning drug tolerance and resistance” is recommended to characterize in more detail the difference between the terms of tolerance and resistance of tumors, and focus on the difference with the term of “therapy effectiveness”, for which there are evaluation criteria. In addition, it would be useful to provide information on existing public databases devoted to the study of drug resistance in oncology.

3.      The section “4. Current treatment strategies may be counterproductive” should contain more up-to-date information on phenotypic plasticity in terms of changing the strategy for approaches to treatment and the selection of predictive markers (for example, works confirming the dedifferentiation of non-stem tumor cells into cancer stem cells (f.e. doi: 10.3389/fphar. 2020.599965; doi: 10.3390/ijms23095058; doi: 10.3389/fimmu.2020.01280).

4.      In my opinion, Figure 2.” Schematic illustration of Waddington's epigenetic landscape" does not allow the reader to get closer to the main idea of the presented review. I recommend replacing this illustration with a more informative one that reflects the main concept of the section.

5.      In conclusion, I would like to see the final illustration summarizing the main theses of this review, as well as identify possible ways to solve the existing problem.

Self-citation - 20%.

I recommend to accept the article for publication with significant corrections.

Author Response

  1. The introduction and conclusion of the article do not agree with each other. The problem identified in the introduction of the review is not fully discussed in the course of the article. The conclusion does not summarize the actual content of the work. In general, in the article I would like to see more facts that are contradictory, and then a summary conclusion of the author of the review.

Author’s response: We appreciate the Reviewer’s comment and have now accordingly discussed the two important papers pointed out by Reviewer #1 (see above) as well as added text in the Introduction and Concluding Remarks sections to eliminate the perceived ‘contradiction’ and to provide a fair and unbiased view to the reader.

  1. The section “3. Discerning drug tolerance and resistance” is recommended to characterize in more detail the difference between the terms of tolerance and resistance of tumors and focus on the difference with the term of “therapy effectiveness”, for which there are evaluation criteria. In addition, it would be useful to provide information on existing public databases devoted to the study of drug resistance in oncology.

Author’s response:

We thank the reviewer for these helpful suggestions. We have now cited the CancerDR drug resistance database, which is an useful source of information on various anticancer drugs and the associated resistance pathways. The distinction between the terms tolerance and resistance is still a grey area in the clinical field. However, the consensus in the literature regarding the use of the term ‘tolerance’ is for temporary survival of cancer cells in presence of a drug, which may revert to sensitivity once the drug is withdrawn. ‘Resistance’ refers to more permanent ineffectiveness of the drug towards the tumor. We have made these distinctions clear in the manuscript. However, we feel that a more detailed discussion on the subject is beyond the scope of this minireview, especially since the concept of drug resistance is not concrete and is still an evolving field.

  1. The section “4. Current treatment strategies may be counterproductive” should contain more up-to-date information on phenotypic plasticity in terms of changing the strategy for approaches to treatment and the selection of predictive markers (for example, works confirming the dedifferentiation of non-stem tumor cells into cancer stem cells (f.e. doi: 10.3389/fphar. 2020.599965; doi: 10.3390/ijms23095058; doi: 10.3389/fimmu.2020.01280).

Author’s response: We appreciate the comment and totally agree with the reviewer. Accordingly, we had text discussing the several important papers [Zheng et al, Front Pharmacol, 2021; Ibragimova et al, the Reviewer pointed out.

  1. In my opinion, Figure 2.” Schematic illustration of Waddington's epigenetic landscape" does not allow the reader to get closer to the main idea of the presented review. I recommend replacing this illustration with a more informative one that reflects the main concept of the section.

Author’s response: We appreciate the Reviewer’s comment and suggestion to replace Figure 2. We have now added two new figures to better illustrate our point of view. In our opinion, Waddington’s landscape is a conceptually elegant way of describing the non-genetic transformations in a cell population and the fact that a common progenitor phenotype can differentiate into alternative states under different environmental conditions. Such changes are actuated by cellular biochemical networks independent of genetic mutations and can facilitate the emergence of temporary drug-tolerant populations. We have elaborated these concepts in the main text and hence would like to retain Fig. 2 as an aide in understanding those discussions.

  1. In conclusion, I would like to see the final illustration summarizing the main theses of this review, as well as identify possible ways to solve the existing problem.

Author’s response: This is an excellent suggestion, and we really thank the Reviewer for bringing this up. We have accordingly added two new figures (Fig. 4 and Fig. 5) that succinctly summarizes the gist of this article.

I recommend to accept the article for publication with significant corrections.

Author’s response: We wish to thank the Reviewer for constructive comments that have helped improve the presentation.

Round 2

Reviewer 2 Report

The revised version of the article has become more demonstrative and scientifically substantiated. Fully agree with the other reviewer's comments. I am sure that the comments made have improved the submitted article. I recommend accepting this version of the manuscript for publication.